# The Renewed Interest on Brunogeierite, GeFe_2_O_4_, a Rare Mineral of Germanium: A Review

**DOI:** 10.3390/molecules27238484

**Published:** 2022-12-02

**Authors:** Marco Ambrosetti, Marcella Bini

**Affiliations:** 1Department of Chemistry, University of Pavia, Viale Taramelli 16, 27100 Pavia, Italy; 2C/o Department of Chemistry, Consorzio per i Sistemi a Grande Interfase (CSGI), Via della Lastruccia 3, 50019 Sesto Fiorentino, Italy; 3National Reference Centre for Electrochemical Energy Storage (GISEL)—Consorzio Interuniversitario Nazionale per la Scienza e Tecnologia dei Materiali (INSTM), Via G. Giusti 9, 50121 Firenze, Italy

**Keywords:** GeFe_2_O_4_, Fe_2_GeO_4_, brunogeierite, physical properties, SIBs, LIBs

## Abstract

GeFe_2_O_4_, also known as brunogeierite, is a rare mineral of germanium. It has a normal spinel structure and, as with many other spinels, amazing functional properties thanks to its peculiar structural features. In the past, its spectroscopic, optical, magnetic and electronic properties were determined; then, for many years, this compound was left behind. Only recently, a renewed interest in this oxide has arisen, particularly for its application in the electrochemical field. In this review paper, the crystal structure of GeFe_2_O_4_ will be described, as well as the synthesis methods required to obtain single crystals or polycrystalline powders. Its spectroscopic, magnetic, optical and electrical properties will be reported in detail. Then, successful applications known so far will be described: its use as anode in Lithium Ion and Sodium Ion Batteries and as electrocatalyst for urea oxidation reaction.

## 1. Introduction

Germanium is an element of subgroup IVa of the periodic table. It has an electrical behaviour that lies between that of a metal and an insulator; some of its compounds are similar to those of non-metals. The possible oxidation states of Ge are +2 and +4, with the latter at the base of the most stable compounds. Germanium ions prefer tetrahedral coordination as silicon, and due to many similarities isostructural compounds of Si and Ge can form. However, differently from silicon, the natural minerals of Ge are few and very rare or even unique to a specific place: the most diffused are sulphides, oxides, sulphates, hydroxides and germanates [1].

In particular, GeFe_2_O_4_ (GFO), whose mineralogical name is brunogeierite, can be found in large amount in Namibia, in the Tsumeb deposits (Figure 1) [2].

The existence of these deposits was explained by the presence of a fracture zone close to the Tsumeb orebody. Oxidation processes caused by meteoric water were at the base of the stabilization of a variety of minerals, famous worldwide for their beauty. In this zone, the primary minerals (germanite Cu_13_Fe_2_Ge_2_S_16_ and renierite (Cu,Zn)_11_(Ge,As)_2_Fe_4_S_16_) were transformed into germanium oxides (mainly brunogeierite GeFe_2_O_4_, otjisumeite PbGe_4_O_9_, and bartelkeite PbFeGe(Ge_2_O_7_)(OH)_2_ H_2_O). Brunogeierite was also found in small amounts in other locations, such as the French Pyrenees and the Marimanti area in Kenya.

A curious aspect concerning GeFe_2_O_4_ was the determination of the right oxidation states of the cations, which was rather difficult. In the updated version of the *IMA List of Minerals* of 2011 and in earlier books, the formula of brunogeierite was written as Ge^2+^(Fe^3+^)_2_O_4_ and its Strunz classification was 4.BB.05 (Oxides, Spinel Group). However, the presence of both Ge^2+^ and Fe^3+^ in the same compound was suspicious and, due to the scarcity and instability in humid air of synthetic compounds containing Ge^2+^ ions, the presence of Ge^2+^ in brunogeierite was considered unlikely [3]. Thus, on the basis of bond-valence calculations, the ideal formula was determined as Fe^2+^_2_Ge^4+^O_4_ [3]. The mineral was classified as a nesogermanate member of the ringwoodite group (Strunz classification 9.AC.15). A new IMA classification for the spinel supergroup minerals included brunogeierite into the ulvöspinel A^4+^B^2+^2O_4_ subgroup, where A and B cations were usually Si^4+^, Ge^4+^, Ti^4+^ and Mg^2+^, Mn^2+^, Fe^2+^ or vacancy, respectively [3].

The ferrite spinels, among which brunogeierite could be numbered, are well-known and commonly studied materials with an impressive range of applications extending from millimeter wave integrated circuitry to power handling, simple permanent magnets and magnetic recording, catalysis, sensors, energy, nanomedicine and imaging in magnetic resonance [4,5,6,7,8,9]. Some examples of the most commonly studied and applied ferrites are represented by ZnFe_2_O_4_, CoFe_2_O_4_, Fe_3_O_4_, NiFe_2_O_4_, MgFe_2_O_4_, and MnFe_2_O_4_ [10,11,12,13,14], each of them having peculiar functional properties.

Therefore, since GFO shares the crystal structure features with most applied spinels, we can foresee similar applications. However, after the publication of some studies many years ago [15,16,17,18,19], interest in GeFe_2_O_4_ vanished for a long time, as demonstrated by the small number of papers published up to now (about 20). Nevertheless, in these last four years, researchers finally recognized the great potential of GFO, which could be exploited in many different fields.

In this review, we will first describe the GFO crystal structure and the synthetic methods employed for obtaining powders, as well as single crystals. Then, the magnetic, optical, spectroscopic and electrical properties, determined up to now will be described. Finally, its recent electrochemical and catalytic applications will be discussed.

## 2. Crystal Structure

The crystal structure of brunogeierite (Figure 2) was determined by Welch in 2001 on a single crystal from Tsumeb [20].

In ref. [20], however, as in other previous studies, nothing was declared about the valence states of the cations in the brunogeierite. The confusion about the right valence states of Fe and Ge ions in GFO probably began from the title of the paper of Fleischer [21] reporting the term “Germanium-Ferritspinell” (Ferrit is, in fact, the common term for the Fe-spinel containing Fe^3+^) [22]. This may have been at the base of the subsequent confusion of the terms *ferrit* and *ferric* and consequently about the valence states of cations. In 2013, on the base of bond-valence calculations, the structure was newly verified and definitively established [3]. GeFe_2_O_4_ is a normal cubic spinel with *Fd-3m* space group and a lattice parameter of 8.4127(7) Å (cell volume of 595 Å^3^). There are eight formula units in the unit cell, as expected for the spinels’ structure. The value for the oxygen coordinate in GFO (Ge and Fe ions are located on special sites) is 0.2466(1) [20]. Taking into consideration that for an ideal cubic close-packing of oxygens the value is 0.25, brunogeierite has a nearly perfect oxygens framework, in which Fe^2+^ occupies half of the octahedral interstices and Ge^4+^ one eighth of the tetrahedral. The Ge-O bond length was 1.771(2) Å, very close to the sum of the ionic radii of the involved ions (0.39Å + 1.38Å = 1.77 Å) [23], but slightly shorter than those reported for other synthetic germanate spinels such as Mg_2_GeO_4_, Co_2_GeO_4_ and Ni_2_GeO_4_ (1.775–1.801 Å). The shorter bond length in GFO can be explained by the presence of some Fe^3+^ on the Ge^4+^ site, as suggested by the results of the bond valence calculations [3]. The Fe-O bond is, instead, 2.132 Å, slightly shorter than the sum of the ionic radii (0.78Å + 1.38Å = 2.16 Å) [23]. In this case, some Fe^3+^ could be present on the octahedral sites, influencing the bond length value. The presence of a small amount of Fe^3+^ ions at both tetrahedral and octahedral sites suggested the formation of a solid solution with the magnetite Fe_3_O_4_ phase [20]. This is not surprising because a partial miscibility of Fe_3_O_4_ and GeFe_2_O_4_ was determined at high temperature by the electromotive force (EMF) method, consisting in measuring the EMF of the oxygen concentration galvanic cell with a solid oxygen conducting electrolyte (ZrO_2_ + Y_2_O_3_); these results were confirmed by electron probe microanalysis [24].

The cubic symmetry of GFO was maintained up to 5 K, without structural distortion, as demonstrated by synchrotron X-ray powder and neutron diffraction measurements [25]. However, by increasing the pressure, it was suggested that, at about 20–22 GPa, a reduction from cubic (*Fd-3m*) to tetragonal (*I4_1_/amd*) symmetry occurred, as determined from the analysis of the band shifts of the Raman spectra and the colour change of crystals [26]. The same authors suggested, on the base of the presence of additional bands in the Raman spectra, a partial inversion of the spinel structure, which at ambient pressure and temperature had normal structure.

## 3. Syntheses

As previously discussed, GeFe_2_O_4_ can be found in nature in peculiar locations in the form of single crystals, which were usually employed for crystal structure determination [20]. However, single crystals were also synthesized in the laboratory and were used, for example, for high pressure Raman studies and for the determination of optical and electrical properties [15,26]. For electrochemical and electrocatalytic applications, GFO was used instead in the form of nano-powders, which are more easily synthesized.

### 3.1. Single Crystals

Strobel et al. [15] grew GFO crystals by the chemical vapour transport method, using TeCl_4_ as transporting agent. In brief, the chosen reagents Fe, Fe_2_O_3_ and GeO_2_ were allowed to reach equilibrium at 920 °C, in a silica tube; then the transport proceeded by decreasing the temperature of the growth zone to 760 °C. The obtained crystals were octahedral with edges up to 4 mm (Figure 3A) and were black coloured, but when powdered turned to light brown, probably due to traces of Fe^3+^ ions, differently from GFO powder, which was black.

In a more recent paper [26], the synthesis of GFO single crystals was carried out in thermo-gradient hydrothermal conditions in an autoclave at about 600–650 °C and 100 MPa with the use of gold-lined inserts. Nutrient materials were stoichiometric amounts of iron wire and germanium oxide, located along the insert length and at the bottom, respectively. After the addition of a solution of 30 wt% of boric acid, the inserts were hermetically sealed, weighed and placed in an autoclave heated in an electric oven up to 650 °C (lower part) and 600 °C (upper part) for 20 days (Figure 3B). The crystals were mainly composed of octahedron {111} faces, with the development of twinning according to the spinel law, while faces of rhombic dodecahedron {110} were rarely present (Figure 3B).

The synthesis of GFO single crystals, even if occurring successfully, is, however, a long and demanding synthesis that was in fact used up to now only to obtain materials for the study of optical, magnetic and spectroscopic properties, or to better analyse crystallographic features.

### 3.2. Powders

As can be inferred from the published papers, the preferred syntheses to obtain polycrystalline GFO were the solid state, hydrothermal and freeze drying. The conventional solid-state synthesis is the oldest method for the synthesis of oxides, since it is economic, efficient and easily scalable. Some disadvantages could be represented by particle agglomeration and growth. The use of mechanical milling during the preparation could guarantee the formation of the desired compounds without additional thermal treatments or at lower temperatures. Some drawbacks could be related to the required highly energetic milling for an extended time and a possible contamination of the milled sample by the materials of balls and jars. However, the tuning of nanoparticles could be easily obtained by changing the milling parameters such as the container, speed, time, ball-to-powder weight ratio and milling atmosphere. For a typical solid-state synthesis [15], stoichiometric amounts of GeO_2_, Fe and Fe_2_O_3_ were mixed, then heated in an evacuated silica tube for 40 h at 800 °C and for other 24 h at 950 °C, with intermediate grindings. Possible traces of magnetite could be present, due to the detection of Fe^3+^ ions. Perversi et al. [25] used a very similar synthesis route with small variations: the same reagents were pressed into a pellet and heated at 900 °C for 60 h, then cooled to room temperature in about 12 h. GFO sample was a pure spinel phase as detected from X-ray Diffraction (XRD) patterns. The mechano-chemical route could represent an improvement of the classical solid-state synthesis, helping the direct formation of the material during milling or allowing its production at low temperature. In the case of GFO, a mixture of α-Fe_2_O_3_, Fe and GeO_2_ in the molar ratio 2:2:3 was milled for different times (up to 2 h) in a planetary ball milling apparatus at room temperature under argon atmosphere [27]. The XRD results are shown in Figure 4A: it is evident that a pure GFO spinel was formed after only 2 h of mixing.

An equally widespread synthesis of GFO was the hydrothermal method [28,29], useful for easily tuning the particle size and shape and providing highly homogeneous products. The main advantage of hydrothermal synthesis is that it happens under non-standard conditions, so that unconventional crystallization pathways can occur. Together with the numerous advantages, a drawback could be represented by the need that the synthesizing compounds are not sensitive to aqueous ambient. In addition, attention should be paid to a proper choice of precipitating agents that can influence the physical and chemical characteristics of the materials.

In a typical procedure [28], FeCl_2_, GeO_2_ and NaOH were used as starting precursors. FeCl_2_ and GeO_2_ were separately dissolved in water (GeO_2_ with the addition of NaOH to obtain the solubilisation). The two solutions were then mixed together, transferred into an autoclave and heated at 180 °C for 24 h. The obtained product, after washing and drying, was treated at 500 °C for 3 h to obtain Fe_2_GeO_4_. For the subsequent use as anode in Lithium-Ion Batteries (LIBs) and Sodium Ion batteries (SIBs) a carbon coating was added by another hydrothermal treatment with glucosamine addition. A very similar synthesis was reported in [30]: the changes concerned the temperature of the heating step in the hydrothermal setup (160 °C) and that of the thermal treatment of the drying step (90 °C).

**Figure 4 molecules-27-08484-f004:**
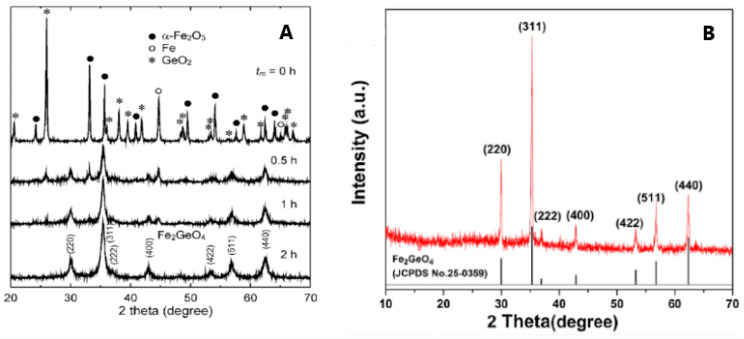
(**A**) XRD patterns of the GFO samples obtained by mechano-chemical synthesis at different mixing times [27], Reproduced with permission, Copyright De Gruyter 2008; (**B**) 3D GFO/N-CNSs XRD pattern [31], Reproduced with permission, Copyright Elsevier 2018.

The freeze-drying technique was used for the preparation of 3D interconnected N-doped ultrathin carbon nanosheets (3D Fe_2_GeO_4_/N-CNSs) with a peculiar morphology to be applied as anodes in LIBs and SIBs. Freeze-drying, usually applied in the food or pharmaceutical industries, even if a somewhat expensive treatment, could provide, in this specific case, the advantage of removing water from the gel by maintaining a favourable structure with the required porosity for the subsequent electrochemical application. The reagents C_6_H_5_O_7_(NH_4_)_3_, FeCl_2_ and NaOH were dissolved in water, the pH was adjusted to 2–3 by HCl, then NaCl was added under magnetic stirring. The solution was frozen in a refrigerator at −20 °C for 24 h to remove the water in the resulting gel. The powder was then annealed at 500 °C for 2 h under Ar, cooled to room temperature and washed with water to remove NaCl [31]. In Figure 4B, the XRD pattern of the composite is shown, demonstrating the formation of the pure GFO spinel phase.

In Table 1, the discussed syntheses (for both single crystals and powders) are summarized for a better comparison and for easier reference.

## 4. Physico-Chemical Properties

### 4.1. Spectroscopic Features

Spectroscopic techniques can provide many hints about the oxidation states of transition metals and their structural features and, specifically for spinel phases, the inversion degree. In addition, they could be used to verify the presence of impurities, particularly iron oxides in ferrite spinels, some of them very difficult to detect, but, if present, altering the intrinsic functional properties of the samples. GFO single crystals and powders were characterized by many spectroscopic techniques, such as Mössbauer, X-ray Photoelectron Spectroscopy (XPS), Infrared Spectroscopy (IR) and Raman. The Mössbauer spectroscopy, being specific on Fe ions, could be useful to determine their oxidation states and coordination, so suggesting the possible presence of impurity phases. This technique was applied for many years on ferrous spinels and on GFO [16,19,26,27]. A typical Mössbauer spectrum of GFO single crystal [26] consists of a symmetric doublet with an isomeric shift (IS) = 1.104(1) mm/s and a quadrupole splitting (QS) = 2.845(1) mm/s, corresponding to Fe^2+^ ions in octahedral sites, confirming the normal spinel structure of GFO. The reported IS values are generally similar in the different papers, while the small differences in QS values could be due to the synthesis methods and the purity of the samples. In Figure 5A, the room-temperature Mössbauer spectra of bulk (prepared by conventional solid state route) and nanoscale mechano-synthesized GFO are compared [27].

The spectrum of the GFO bulk (Figure 5A, curve a) consisted of a symmetrical doublet with IS = 0.96(4) mm/s and QS = 2.75(4) mm/s characteristic of octahedrally coordinated Fe^2+^ ions, as determined on single crystals by Setkova et al. [26], confirming the normal spinel structure of GFO. The nanocrystalline sample presented instead an asymmetric doublet (Figure 5A, curve b), that was fitted by two sub-spectra: the first one due to Fe^2+^ ions in B sites (IS = 0.99(6) mm/s, QS = 2.7(2) mm/s), while the other to Fe^2+^ ions in A sites (IS = 0.61(6) mm/s, QS = 1.2(3) mm/s) of the AB_2_O_4_ structure. This suggested the stabilization of an inverted spinel, with an inversion degree of about 0.67, i.e., a nearly random distribution of cations with maximum configurational entropy.

X-Ray Photoelectron Spectroscopy is a powerful surface technique for the identification of elements and their valence states as well as to point out their possible chemical bonds. The collection of the survey spectrum could demonstrate the presence of the expected elements, while high-resolution spectra on single elements are suitable to confirm valence states and environments. A typical GFO XPS survey spectrum is reported in Figure 5B [28], showing the expected Ge, Fe and O elements. The high-resolution spectra were usually collected on Ge and Fe ions, but also on O or C (if present). The Fe 2p spectrum was in general constituted by two characteristic peaks corresponding to the core level spectral lines of Fe 2p_3/2_ and Fe 2p_1/2_, at about 711.0 and 725 eV, typical of Fe^2+^ ions. In some cases, peak broadening (such as reported in [30]) or satellites can suggest the presence of a small number of Fe^3+^ ions. The presence of germanium as Ge^4+^ ions is demonstrated by collecting the 3d_5/2_ spectrum (peak at about 31.7 eV) [28] or the 2p_3/2_ spectrum (peak at about 1220 eV) [30]. XPS is also useful for samples containing carbon, in the form of coating layer or as composites, because the deconvoluted spectra of carbon could provide significant information. For example, in the paper of Subramanian et al. [28], the peaks at 284.4 and 286.6 eV of the C1s spectrum were attributed to the C-C bond in aromatic rings and C-O group, respectively. The C-O bond is formed at the interface between the carbon shell and GFO, so facilitating electron transfer and justifying the good electrochemical performances.

The Infrared spectrum of GFO was reported many years ago together with those of other germanates [18] and it is constituted by three bands at 688, 402 and 319 cm^−1^. Assuming that GFO is a normal spinel, as previously demonstrated by other spectroscopic techniques, the peak at 688 cm^−1^ could be attributed to the asymmetric stretching vibration of GeO_4_ tetrahedra. The other two modes were more difficult to rightly assess: the band at 402 cm^−1^ was probably due to the deformation of GeO_4_ tetrahedra, while that at 319 cm^−1^ could be due to FeO_6_ octahedra. This explanation could also be reversed, i.e., the first band could be due to FeO_6_ and the other to GeO_4_. In that paper, a sure explanation was not provided and, up to now, no IR patterns have been published on this material to clarify these doubts.

Raman spectroscopy, providing similar or complementary information with respect to IR, was more often applied to GFO, both at ambient and high pressure. The aim of the Raman studies, in general, was to characterize the spinel phase structure and, in case of carbon coating, to have information about the ordered or disordered state of carbon, for which Raman is a phenomenal probe [26,28,29]. The factor group analysis for Brunogeierite, with *Fd-3m* space group, suggested 42 vibration modes, 3 acoustic and 39 optical modes. However, only five Raman active vibration modes were allowed for spinel oxides, i.e., the A1g + Eg + 3F2g. The Raman spectrum of synthetic brunogeierite (Figure 6A, together with those of other germanium spinels) shows an intense band at 756 cm^−1^ and other lower bands at about 644, 472, 302 and 205 cm^−1^ [26].

The strongest peak was assigned to the A1g mode and that at 644 cm^−1^ to F2g, by analogy with ringwoodite, and could possibly be due to GeO_4_ tetrahedra. From the factor group analysis, it seemed that only the vibration modes related to the GeO_4_ tetrahedra should be present and the modes should not significantly vary with the substitution of Fe^2+^ with other cations (as demonstrated in Figure 6A). The other F2g modes were located at 472 and 205 cm^−1^, while the 302 cm^−1^ band was due to the Eg mode. It seemed that the bands at 302 and 205 cm^−1^ were due to stretching vibrations of the Fe-O bond. The high-pressure Raman spectra of GFO are shown in Figure 6B [26]. The five main bands detected in the ambient pressure spectrum (Figure 6A) were again present but shifted to higher wavenumbers, due to the reduction of interatomic distances. Some bands shifted to higher wavenumbers and decreased in intensity and others split into at least two components (Figure 6B). The colour of the crystal changed at 22.7 GPa from brown-orange to reddish and became opaque black up to 30.2 GPa (Figure 6B). The same piezo-chromic effect was verified for other Fe^2+^ minerals and was due to changes in the electronic band gap caused by crystal structure distortion and transition of octahedral Fe^2+^ from the low-spin (LS) to high-spin (HS) electronic state [32,33]. Some extra bands also appeared at high pressure (red lines in Figure 6B), that, by analogy with the studies of other spinels, could be explained by different reasons: (i) formation of dissociation products; (ii) coexistence of the cubic phase with polymorphs; (iii) cations disordering and iv) distortion of crystal structure. It seemed that GFO does not decompose below 30 GPa and does not coexist with polymorphs, so the additional bands may be due to partial spinel inversion. The difference in ionic radii of Fe^2+^ and Ge^4+^ ions inhibited the inversion in single crystals but not in nanocrystalline GFO that, when obtained from mechano-synthesis showed an inversion degree of 0.67 [27]. The new bands and the band splitting can be due also to structure distortion towards the tetragonal structure (*I4_1_/amd*), the most probable distortion of the cubic spinel containing Fe^2+^ ions.

### 4.2. Magnetic Properties

The magnetic properties of GFO were measured, for the first time, on polycrystalline samples by Blasse et al. in 1963 [17], in the range 90–900 °K. The high-temperature susceptibilities followed the Curie-Weiss law: the values of the Curie temperature θ_A_, the effective Bohr magnetons number n_eff_ and the Neel temperature T_N_ were −15°, 5.40 (in μ_B_) and 10 °K, respectively. The n_eff_ strongly deviated from the expected 4.90 spin only value, suggesting the presence of spin-orbit coupling, so the θ_A_ value cannot be considered a measure of magnetic interactions, because the magnetic moment changes in the low-temperature region. The results of Blasse et al. [17] were confirmed, more recently, on synthetic single crystals of GFO (grown as explained in par. 3.1) [15]. A nonlinear variation of χ^−1^ vs. T was found below 110 °K. The determined θ_A_ and n_eff_ values were −25 °K and 5.26: due to the spin-orbit coupling of Fe^2+^ ions, n_eff_ was related to the trigonal field splitting of the t2p levels of cations in B sites of about 950 cm^−1^. The difference with respect to the value of 600 cm^−1^ of Blasse et al. [17] may be due to the lower Fe^3+^ content in the powdered single crystal sample.

The magnetic properties of polycrystalline GFO sample were recently measured in a more accurate way, to determine the low temperature spin order [25] and a frustrated itinerant spin model for spinels by using a detailed numerical study [34]. The magnetic results were interpreted also thanks to the use of neutron powder diffraction data [25]. GFO magnetic susceptibility measurements (Figure 7a) showed two magnetic transitions with a susceptibility maximum at T ≈ 9 K and divergence of field and zero-field cooled curves at T ≈ 7 K (see inset in Figure 7a).

Alternating Current (AC) measurements showed no frequency dependence in the low-temperature range, suggesting the absence of spin–glass behaviour (Figure 7b). A broad magnetic contribution to the low-temperature heat capacity extended to around 50 K (Figure 7c), but the integrated entropy over the two transitions (5.77 J mol^−1^ K^−1^ per Fe^2+^) was only 43% of the theoretical value (Rln5) expected for long-range order of S = 2 spins. The synchrotron X-ray diffraction powder data at 5 K, as well as the neutron data, showed that the crystal structure does not change at low temperatures (Figure 7d). This was unexpected, because the spin orders in oxide spinels caused, in general, lattice distortions, as for ZnV_2_O_4_, LiMn_2_O_4_, MgCr_2_O_4_ and Co_2_GeO_4_. The measurements of GFO indicated that the orbital states and a large fraction of Fe^2+^ spins remained dynamic below the two magnetic transitions. From the observation of neutron diffraction patterns, sharp magnetic peaks appearing below 9 K with an additional weak peak below 7 K, suggested long-range spin order (Figure 7e). From these numerous and accurate data, it was demonstrated that the single B octahedral position was split into two magnetically distinct Fe1 and Fe2 sites. From the fits, the maximum resultant amplitude was μ = 4.05 μ_B_, in agreement with the ideal value of 4 μ_B_ for high-spin Fe^2+^. The average ordered moment magnitude was 64% of the ideal value, so about one-third of the spins remained dynamic below the magnetic ordering transitions. This behaviour was unusual in non-metallic materials: the reason was attributed to a frustration wave order in GFO derived from exchange interactions between the ordered spin components in a sublattice via the dynamic components of their neighbours in the other.

The peculiar magnetic features of GFO are not shared by all the Ge based spinels, because being Ge^4+^an inactive ion, they are determined by the B^2+^ sublattice; so a variety of different behaviours could be identified. For example, GeCo_2_O_4_ is a frustrated spinel that orders antiferromagnetically at T_N_ of about 20 K–23 K. The magnetic frustration in GeCo_2_O_4_ resulted from factors such as the geometrical frustration of the lattice of the Co^2+^ spins along with the presence of several interlayer antiferromagnetic exchange couplings among the different kind of spins, in addition to the dominant in-plane ferromagnetic exchange coupling [35]. GeCu_2_O_4_ is a little studied spinel due to the difficulty in the synthesis, favoured by high pressure. It has a tetragonally distorted spinel structure in which the octahedral *B* site is occupied by the Cu^2+^ Jahn-Teller active ions, with a local coordination that can be described as almost square planar. Thus the structure is comprised of alternating mutually perpendicular layers of 1D CuO_2_
*S* = 1/2 chains interconnected via GeO_4_ tetrahedra. The 1D nature of magnetic interactions within the *S* = 1/2 chains is supported by the magnetic susceptibility data that indicate the onset of long-range magnetic ordering at 33 K. Despite the large tetragonal distortion of the pyrochlore sublattice, the magnetic interaction between Cu ions in GeCu_2_O_4_ remains frustrated, in analogy to the Cs_2_CuCl_4_ case [36]. Finally, GeNi_2_O_4_ has a cubic spinel structure and is a three-dimensional *S* = 1 frustrated magnet with an unusual two-stage transition to the two-dimensional antiferromagnetic ground state. The Ni^2+^ ions, electronically non-degenerate, indicate that the frustration cannot be relieved via structural distortion driven by the cooperative Jahn-Teller effect, similarly to GFO [36]. Therefore, for the Ge spinel oxides, having such a wide diversification of magnetic properties, we can foresee application in different fields.

### 4.3. Optical and Electrical Properties

In 1980, Strobel et al. [15] first studied the optical and electrical properties, together with magnetic properties, of GFO single crystals, obtained as described in par. 3.1. The optical absorption coefficient of GFO was measured in the range 500–1200 nm (Figure 8A), while conductivities ranging from 10^−8^ to 10^−4^ Ω^−1^ cm^−1^ were determined in the temperature range 200–320 °K (Figure 8B).

The broad band at 980 nm was attributed to the ^5^Γ_5_ + ^5^Γ_3_ crystal-field transition of the Fe^2+^ ions in octahedral coordination. The sharply rising absorption edge at 550 nm suggested the onset of a band-to-band transition. The value of the optical band gap was E_B_ > 2.3 eV, the resistivity about 2·10^5^ Ω cm and the activation energy 0.40 eV. The positive value of the Hall constant suggested a p-type conduction. Conductivity and Hall effect measurements showed that the acceptor ionization energy was 0.39 eV and that the mobility was low and independent of temperature, probably due to a narrow valence band and/or large polaron formation. GFO differed, however, with respect to other hopping-type spinels, because its electronic properties were similar to those of NiO at high temperatures.

## 5. Applications

### 5.1. Anode for LIBs and SIBs

Spinels were widely used for electrochemical applications, mainly as anodes in lithium and sodium ion batteries, due to the intriguing features of their cubic crystal structure. So far, the most commonly used spinels are ZnFe_2_O_4_, CoFe_2_O_4_, NiFe_2_O_4_, but recently also GFO was proposed as anode [28,29,31].

The spinels had an insertion mechanism that involved both conversion and alloying reactions, with both lithium and sodium ions, that allowed high-capacity values. The advantages of GeFe_2_O_4_ for use in batteries, if compared with other Ge-based ternary oxides, were the low cost, environmental friendliness, and the abundance of iron.

The lithium intercalation in GFO was drawn as in the following Equations (1)–(4) [31]:Fe_2_GeO_4_ + 8Li^+^ + 8e^−^→ 2Fe + Ge + 4Li_2_O(1)
2Fe + 3Li_2_O ↔ Fe_2_O_3_ + 6Li^+^ + 6e^−^(2)
Ge + 2Li_2_O ↔ GeO_2_ + 4Li^+^ + 4e^−^(3)
Ge + 4.4Li^+^ + 4.4e^−^↔ Li_4_._4_Ge(4)

During the first discharge, GFO converted to Ge, Li_2_O and Fe due to the irreversible conversion reaction (1). The matrix consisting of Li_2_O and Fe can buffer the volume expansion of Ge particles during the alloying/de-alloying process. Furthermore, Fe metal can enhance the electron conductivity, thus leading to better electrochemical performances. Iron and germanium were then involved in a conversion reaction with Li_2_O towards Fe_2_O_3_ and GeO_2_ ((2) and (3)). Finally, Ge metal was lithiated, forming the alloy with the maximum composition Li_4_._4_Ge (4)

For SIBs, an analogous mechanism was hypothesized (Equations (5)–(7)) [31]:Fe_2_GeO_4_ + 8Na^+^ + 8e^−^→ 2Fe + Ge + 4Na_2_O(5)
2Fe + 3Na_2_O ↔ Fe_2_O_3_ + 6Na^+^ + 6e^−^(6)
Ge + xNa^+^ + xe^−^↔ Na_x_Ge      x = 1(7)

Alongside the advantages previously listed, GFO had, however, some insidious issues, in particular the low conductivity and high-volume expansion during cycling, two severe problems to be solved for the implementation of electrodes in practical applications. The mechanical stress induced by the volume changes caused the pulverization of GFO particles and the peeling off from collectors, thus causing a short cycling life, while the low-conductivity of GFO worsen the performance at high rates. The nano-structuring of powders and the formation of hybrid structures were considered as two winning strategies to overcome these issues. In fact, nanoparticles can alleviate the mechanical stress caused by the volume changes of active materials and also shorten the diffusion paths of electrons and ions. The hybrid structures, composed by the active material and conductive carbonaceous species, can enhance the electron conductivity and buffer the volume expansion of active materials, thus improving cycling and rate capability. Currently, the integration of both the strategies in an electrode for the improvement of the electrochemical performances was also challenging.

Subramanian et al. [28,29] coated GFO with carbon through a hydrothermal method and applied it as anode in SIBs. This Fe_2_GeO_4_@C anode showed good discharge capacity, cycling stability and rate capability. In Figure 9A, the discharge capacities of GFO and GFO@C are compared.

The effect of carbon coating was clearly evident when increasing the current density, because the carbon layer provided a continuous electronic pathway between GFO particles.

The positive effect of carbon, which could accommodate volume changes during cycling, was also evident in the capacity values of the long cycling of GFO@C (Figure 9B): an initial discharge capacity of 423.0 mAhg^−1^ and a stable cycling behaviour was shown. After 100 cycles, GFO@C delivered a discharge capacity of 376.5 mAhg^−1^, corresponding to a capacity retention of 89.0%, higher with respect to GFO alone (65%) [28].

The electrochemical reaction mechanism of sodiation-desodiation of GFO was investigated through ex-situ XRD patterns at different states of charge, to verify the mechanism proposed in Equations (5)–(7) (Figure 10).

During the 1st sodiation at 0.55 V (Figure 10, curve b), new peaks appeared at 41.5° and 41.8° (with respect to the pristine electrode, Figure 10, curve a), which can be ascribed to FeO and α-Fe (Equation (5)). The peaks of GFO completely disappeared at 0.01 V (Figure 10, curve c), and the new peaks at 38.7° and 55.8° were due to NaF, deriving from the decomposition of fluoroethylene carbonate (FEC), an additive of the electrolyte. The alloying reaction of Ge was not observed, because the alloy could be amorphous. After de-sodiation at 1.0 and 3.0 V (Figure 10, curves d and e), Fe was oxidized to Fe_2_O_3_ (Equation (6)). After full sodiation at 0.01 V (Figure 10, curve f), Fe metal formed, due to the reversible reaction of Fe_2_O_3_. To demonstrate the practical application of the obtained GFO, a sodium-ion full cell was assembled, with GFO@C as anode and NaCo_0_._5_Fe_0_._5_O_2_ as cathode [28]. An initial discharge capacity of 311.3 mAhg^−1^ was obtained with coulombic efficiency (CE) higher than 99.3%. The results clearly demonstrated that the GFO@C composite could be a promising anode material for SIBs.

Another efficient strategy to overcome the issues of GFO anode was the production of interconnected N-doped ultrathin carbon nanosheets anchored with ultrasmall Fe_2_GeO_4_ nanodots (3D GFO/N-CNSs) by high-temperature calcination method [31]. Its microstructure was investigated by Transmission Electron Microscopy (TEM) (Figure 11a,b), demonstrating the complex nature of the hybrid. This nanostructure provided the synergistic advantages of nanostructured GFO and conductive carbon matrix, with robust bonds between GFO and carbon nanosheets that could accommodate the volume expansion. The electrode was satisfactorily applied as anode for both LIBs and SIBs.

High-resolution TEM (HRTEM) image (Figure 11c) showed that the thickness of the carbon nanosheets was about 3 nm and confirmed that GFO nanodots (black dots) were homogeneously distributed (Figure 11d). Their nature was confirmed from Figure 11e showing a spacing of lattice planes of 0.25 nm, in agreement with the (*440*) plane of GFO. The average size of GFO particles was about 4.6 nm (Figure 11f) and the elemental maps showed that C, N, Ge, Fe, and O elements were uniformly distributed in the powder (Figure 11g). The same analysis performed on pure GFO demonstrated agglomeration of particles (sizes > 500 nm). The formation of stable 3D carbon networks was attributed to the space-confined and self-assembly effect of NaCl used during the synthesis and the strong interfacial bonds between GFO and carbon.

The cycling performances of 3D GFO/N-CNSs in LIBs were compared to those of pure GFO and carbon (Figure 12A,B) [31].

The initial specific discharge and charge capacities at a current density of 0.1 Ag^−1^ were 1697.7 mAhg^−1^ and 1169.8 mAhg^−1^, with a CE of about 68.9%, due to the irreversible decomposition reaction of GFO and the formation of SEI. The CE quickly increased to 94.3% at the third cycle but was about 99.4% during the subsequent cycling. In contrast, the performances of 3D N-CNSs and GFO electrodes were unsatisfactory (Figure 12A). The rate performances of the three electrodes at different rates were compared in Figure 12B. 3D GFO/N-CNS showed outstanding rate performance and cycling stability at different current densities with a reversible specific capacity of about 650 mAhg^−1^ at 6.4 Ag^−1^, much higher than the 372 mAhg^−1^ of graphite. Pure GFO and 3D N-CNSs had instead poor rate performance. These results demonstrated that the unique hybrid nanostructure can enhance electron conductivity and also restrict the aggregation and volume fluctuation of GFO during the charge/discharge process.

Hybrid nanostructured and monodispersed GFO nanoparticles anchored on reduced graphene oxide were synthesized via hydrothermal method and applied as anode in LIBs [37]. The capacities at different current densities ranged between 980 and 340 mAhg^−1^ and a good reversibility was maintained with a CE over 98%. This nanostructure delivered a high reversible capacity of 980 mAhg^−1^ for 175 cycles with a capacity retention of 100% in comparison with that of the second cycle (982 mAhg^−1^) [37]. Again, the excellent cycling performances should be ascribed to the superior material structure of the hybrid.

### 5.2. Electrocatalyst

GFO was also recently used as electrochemical catalysts for the urea oxidation reaction (UOR) [30]. Urea was utilized as an additive to reduce the constant thermodynamic potential and enhance the efficiency of H_2_ generation. Because urea can be oxidized at a more negative potential with respect to that of H_2_O, the UOR can substitute for the O_2_ evolution reaction (OER). However, the UOR has intrinsically slow kinetics due to the six-electron transfer process and the complicated gas evolution. To improve the efficiency of urea electrolysis, various metal oxide electrocatalysts have been investigated due to their low cost, easy synthesis, and high stability under alkaline conditions [38]. The most commonly used catalysts for UOR are Ni-based materials, whose excellence relies on their transformation to NiO and NiOOH as active sites for UOR catalysis. The possible incorporation of other metallic elements can improve the modulation of the electronic structures of Ni active sites [38].

GFO was employed as a bimetallic oxide electrocatalyst to enhance the hydrogen generation using urea-containing wastewater; its performances were compared to those of Co_2_GeO_4_, a spinel phase with manifold applications. Both GFO and Co_2_GeO_4_, whose particle sizes were 150 and 900 nm respectively, were prepared via hydrothermal method and used as anodes for urea oxidation as a counter reaction of the hydrogen evolution reaction (HER) for H_2_ production. In an alkaline electrolyte with urea, the GFO electrode showed a lower potential (1.53 V vs. reversible hydrogen electrode (RHE)) and a smaller Tafel slope (76 mV dec^−1^) than Co_2_GeO_4_ (1.65 V vs. RHE, 79 mV dec^−1^) to reach a current density of 50 mA cm^−2^, implying that GFO reduced the input potential to generate H_2_. These superior performances were attributed to the higher oxidation states of the metal cations, higher electrochemical active surface area, superior surface accessibility, and lower charge transfer resistance with respect to Co_2_GeO_4_. The GFO performances are also superior for example to those of nickel phosphide nanoflake arrays on carbon cloth Ni_2_P NF/CC (1.35 V for 50 mAcm^−2^), Rh-Ni electrode (1.4 V for 50 mAcm^−2^) CoS_2_ nanoneedle array grown on Ti mesh CoS_2_ NA/Ti (1.59 V for 10 mAcm^−2^) and Zn_0_._08_Co_0_._92_P nano-wall array on titanium mesh (1.38 V at 10 mA cm^−2^) [38] and references therein. Therefore, we can conclude that GFO could be considered a valid and competitive electrocatalyst for UOR.

## 6. Summary and Outlook

Researchers are incessantly looking for increasingly performing materials, with peculiar functional properties, allowing possible application in many fields. The AB_2_O_4_ spinels fully meet these requirements thanks to their intriguing structural features. The ferrite spinels, in particular, well-known and characterized materials, have in fact an impressively wide range of applications from simple permanent magnets, catalysis, sensors, energy and nanomedicine to the imaging in magnetic resonance. The GeFe_2_O_4_ brunogeierite ferrite spinel, currently one of the less commonly studied spinels, but sharing its main structural features with the most diffused ferrite spinels, could be possibly proposed for analogues application. In the past, its magnetic, optical and electrical features were determined, even if only few papers were published. This could be due in part to the difficulty in finding suitable natural crystals, which were very rare and located in a few peculiar locations in the world. However, crystals and powders are nowadays easily obtained from different synthetic routes, opening the doors towards a renewed interest on GFO. Recently, thanks to the advantages provided by the spinel structure for lithium/sodium intercalation allowing a double mechanism of conversion and alloying, GFO was proposed as an anode for LIBs and SIBs, with encouraging results, particularly thanks also to nano-structuring and the hybrids’ compounds with carbon. Another promising application seemed to be its use as anode for urea oxidation as a counter reaction of the hydrogen evolution reaction for the efficient H_2_ production.

Due to the intriguing features of GeFe_2_O_4_, we hope that a wider application in other unexploited fields can be foreseen soon, similarly to the most widely diffused spinels.

## Figures and Tables

**Figure 1 molecules-27-08484-f001:**
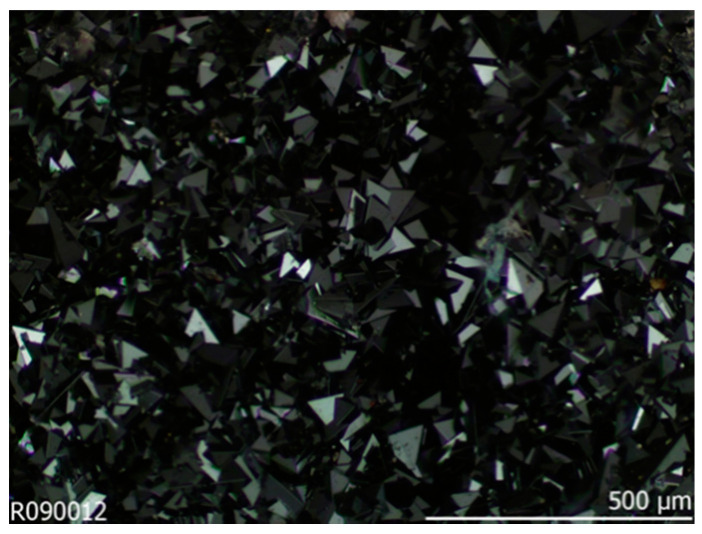
Image of Brunogeierite crystals from Tsumeb, Namibia, taken from RRuff Database [2].

**Figure 2 molecules-27-08484-f002:**
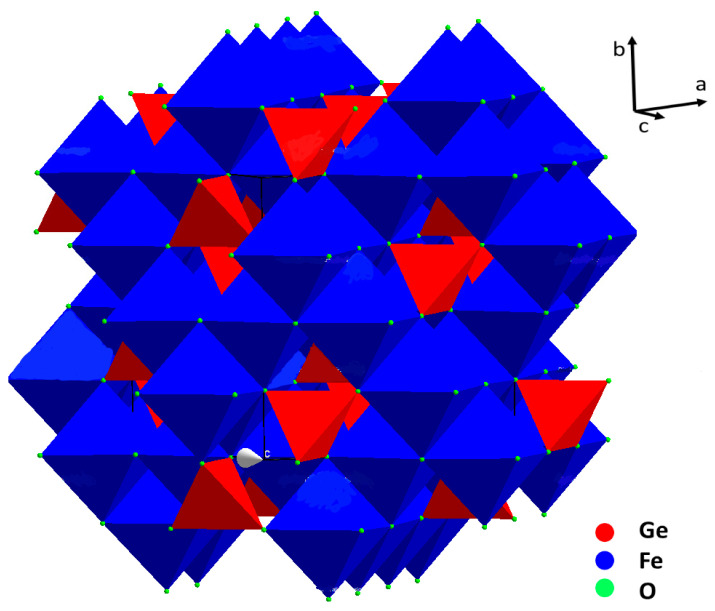
Representation of the brunogeierite crystal structure: FeO_6_ octahedra (blue) and GeO_4_ tetrahedra (red).

**Figure 3 molecules-27-08484-f003:**
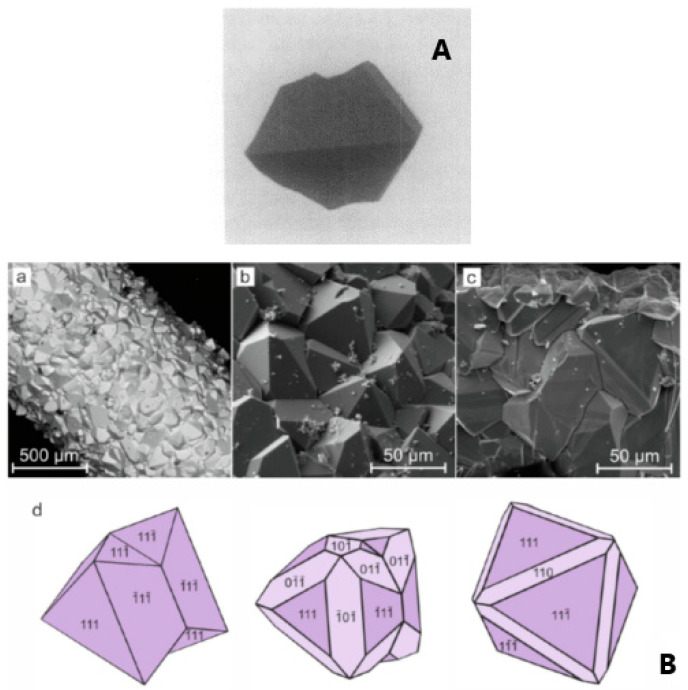
(**A**) Monocrystalline GFO grew by chemical vapour transport method [15]. Reproduced with permission, Copyright Elsevier 1980; (**B**) Scanning Electron Microscopy (SEM) images of brunogeierite crystals: (a) brunogeierite crystals formed on the surface of iron wire; (b, c) enlarged images of crystals druses; (d) idealized morphology and spinel law twinning of brunogeierite crystals [26]. Reproduced with permission, Copyright Elsevier 2022.

**Figure 5 molecules-27-08484-f005:**
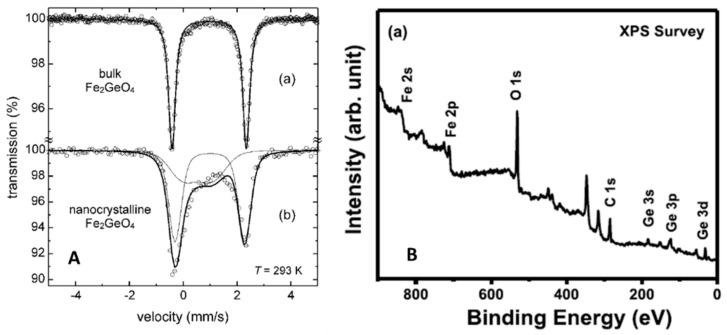
(**A**) Mossbauer spectra from bulk and nanocrystalline GFO [27]. Reproduced with permission, Copyright De Gruyter 2008; (**B**) XPS survey spectrum of polycrystalline GFO [28]. Reproduced with permission, Copyright Elsevier 2019.

**Figure 6 molecules-27-08484-f006:**
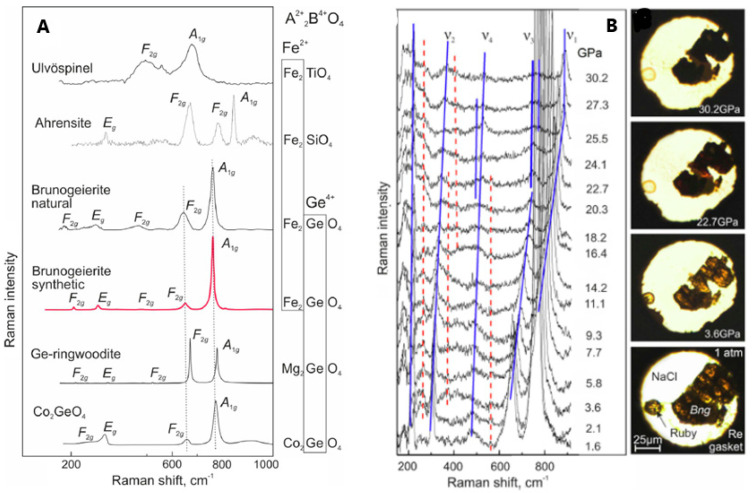
(**A**) Raman spectra of Fe^2+^ and/or Ge^4+^ AB_2_O_4_ spinels; (**B**) Raman spectra of synthetic brunogeierite single crystal at pressures up to 30 GPa (blue lines—shifting trends of the main bands, red dotted lines—shifting trends of the extra bands) and images of brunogeierite crystal (Bng) under compression [26]. Reproduced with permission, Copyright Elsevier 2022.

**Figure 7 molecules-27-08484-f007:**
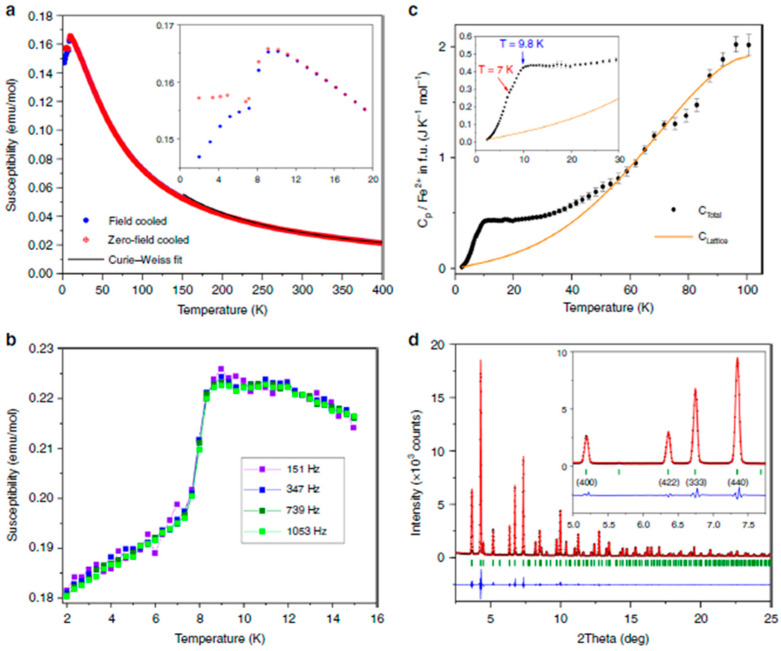
(**a**) Magnetic susceptibility in an applied field of 0.5 T (inset showing the low-temperature region); (**b**) Real part of the AC susceptibility in an oscillating magnetic field with amplitude 9 Oe at different frequencies; (**c**) Heat capacity variation with the lattice contribution fit. The inset shows the low-temperature region (error bars are standard deviations); (**d**) Fit of the synchrotron X-ray diffraction profile at 5 K; (**e**) Magnetic scattering profiles from neutron diffraction data at different temperatures. *hkl* labels correspond to magnetic satellite reflections with different propagation vector [25]. Reproduced with permission, Copyright Springer Nature 2018.

**Figure 8 molecules-27-08484-f008:**
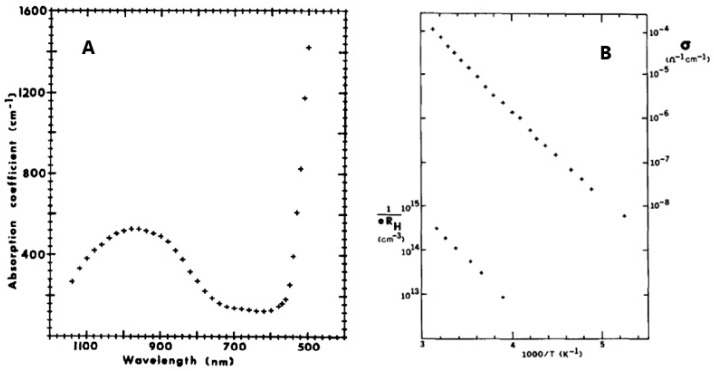
(**A**) Optical absorption spectrum of GFO single crystal at room temperature; (**B**) Conductivity (+) and hole concentration (●) of GFO single crystal as a function of temperature. Ref. [15] Reproduced with permission, Copyright Elsevier 1980.

**Figure 9 molecules-27-08484-f009:**
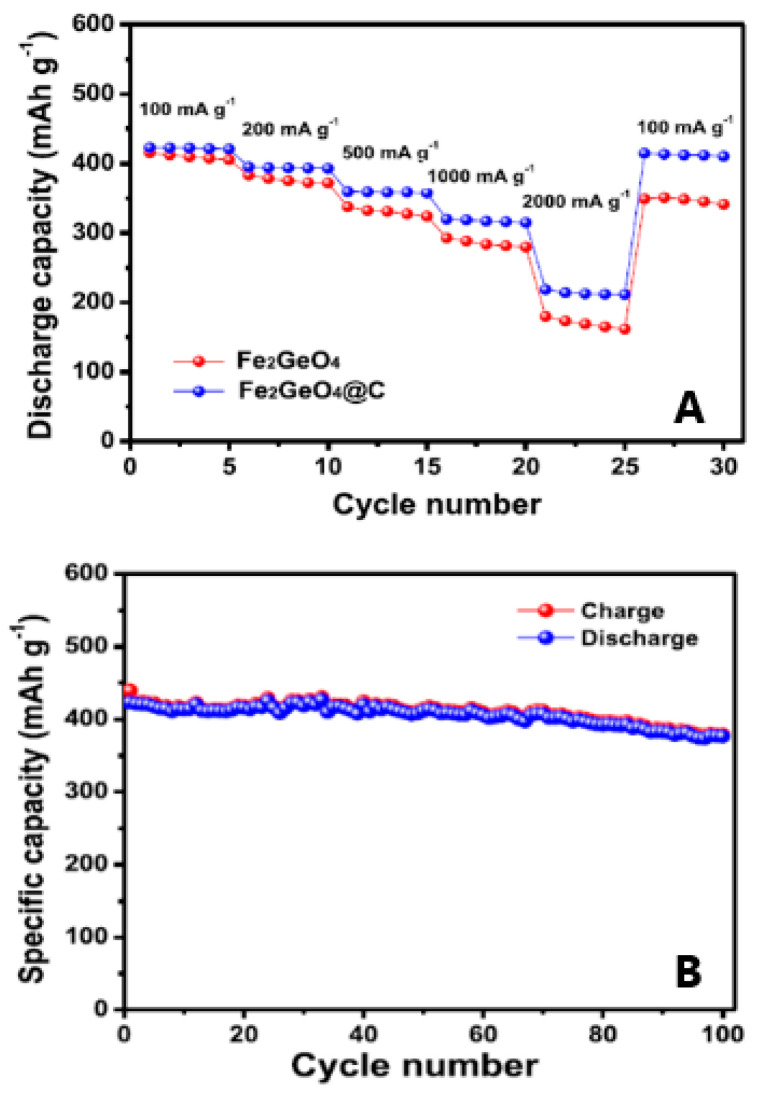
(**A**) Discharge capacities of GFO and GFO@C electrodes with increasing C rate; (**B**) Charge and discharge capacities of GFO@C electrode as a function of the cycle number (current rate: 100 mA g^−1^) [28]. Reproduced with permission, Copyright Elsevier 2019.

**Figure 10 molecules-27-08484-f010:**
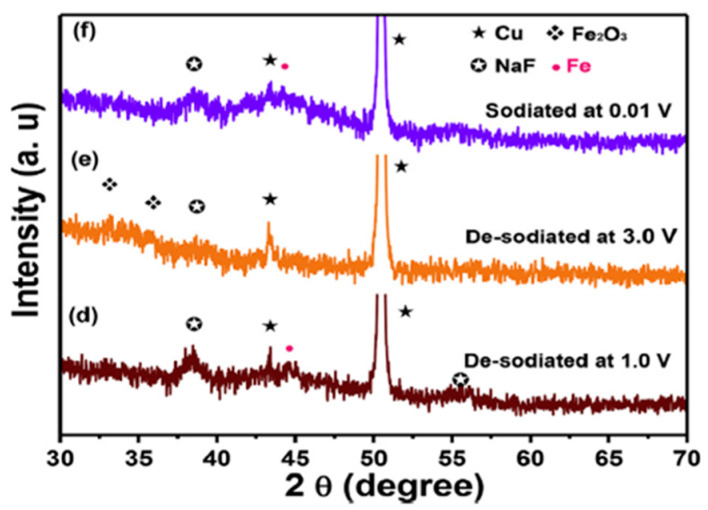
Ex-Situ XRD patterns of GFO electrode at different states of charge: (**a**) pristine electrode, (**b**) sodiated at 0.55 V, (**c**) sodiated at 0.01 V, (**d**) de-sodiated at 1.0 V, (**e**) de-sodiated at 3.0 V, and (**f**) sodiated at 0.01 V [28]. Reproduced with permission, Copyright Elsevier 2019.

**Figure 11 molecules-27-08484-f011:**
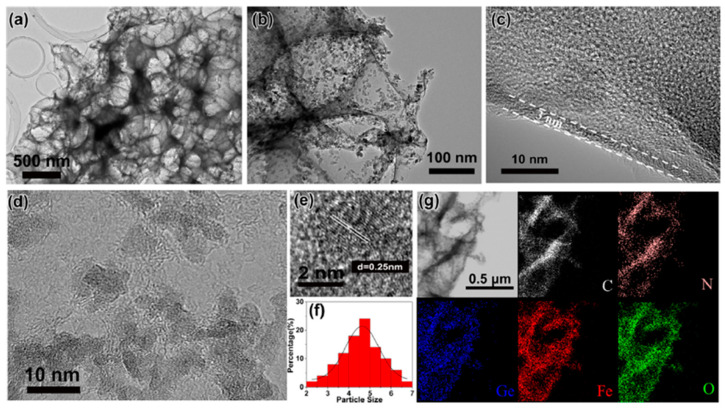
(**a**,**b**) TEM and (**c**–**e**) HRTEM images of 3D GFO/N-CNSs; (**f**) size distribution diagram of GFO nanodots; (**g**) EDS mapping of the elements of 3D GFO/N-CNSs [31]. Reproduced with permission, Copyright Elsevier 2018.

**Figure 12 molecules-27-08484-f012:**
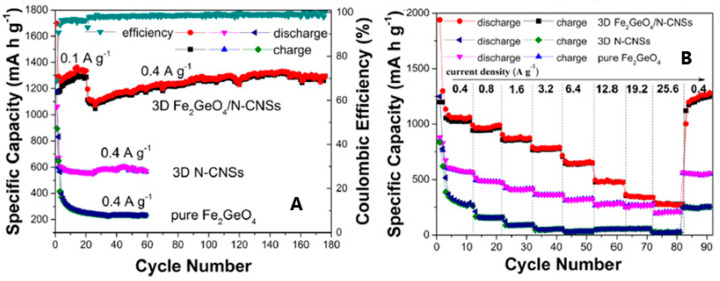
Cycling (**A**) and rate performances (**B**) of 3D GFO/N-CNS compared to those of 3D N-CNSs and pure GFO anodes for LIBs [31]. Reproduced with permission, Copyright Elsevier 2018.

**Table 1 molecules-27-08484-t001:** The main synthesis methods of GFO reported up to now in the literature and described in the text.

Single Crystals				
Synthesis	Reagents	Temperature/Time	Purity	Refs.
Chemical vapour transport method	TeCl_4_ (transporting agent); Fe, Fe_2_O_3_ and GeO_2_	920–760 °C; 11–20 days	-	[15]
Thermo-gradient hydrothermal conditions	Fe wire, GeO_2_, a solution of 30 wt% of boric acid	600–650 °C; 20 days	-	[26]
Powders				
Solid-state	GeO_2_, Fe, Fe_2_O_3_	40 h at 800 °C and 24 h at 950 °C	Traces of Fe_3_O_4_	[15]
Solid state	GeO_2_, Fe, Fe_2_O_3_	Pellet heated at 900 °C for 60 h, and cooled to room temperature in 12 h	Pure	[25]
Mechano-chemical	α-Fe_2_O_3_, Fe, GeO_2_	Milling at 600 rpm in WC jars up to 2 h	Pure only after 2h of milling	[27]
Hydrothermal	FeCl_2_, GeO_2_ and NaOH in water	180 °C for 24 h (or 160 °C for 24h) in autoclave. Then heating at 500 °C for 3 h	Pure	[28,29,30]
Freeze-drying	C_6_H_5_O_7_(NH_4_)_3_, FeCl_2_ and NaOH in water (pH 2–3 by HCl), NaCl	Solution frozen at −20 °C for 24 h. Annealing at 500 °C for 2 h under Ar, washing with water to remove NaCl	Pure	[31]

## Data Availability

Not applicable.

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
