# Peer review of "The Renewed Interest on Brunogeierite, GeFe2O4, a Rare Mineral of Germanium: A Review"

_molecules, 2022, doi:10.3390/molecules27238484_

Round 1
Reviewer 1 Report
The authors present a well-balanced summary of the recent research on Brunogeierite based on its structural properties, synthetic approaches, experimental characterization and applications. The review is supported by a nice overview of the state-of-the-art in the field, with the proper identification of the main experimental achievements and ends with some possible directions to move forward. I personally like very much the style and I consider the review very appropriate due to the novelty and the increasing attention that the material is attracting. I would therefore suggest an acceptance for MDPI Molecules as is.
Minor point:
- Fig. 2: the axes and legend are very small.
Reviewer 2 Report
This review is focused on GeFe2O4 spinel. While most of the B-site spinel undergoes magnetostructural transition, this compound and GeNi2O4 do not seem to undergo a structural transition to relax the frustration. Since the Fe compound is not known for any exciting properties, I suggest discussing other transition metals like GeNi2O4, GeCo2O4, GeCu2O4, and so on.
Minor corrections: In line 327, From the observation of the X-ray diffraction pattern, sharp magnetic peaks... Here it should be neutron diffraction and not X-ray diffraction
Reviewer 3 Report
This article gives a general overview of the crystal structure, synthesis methods, some properties, and mainly applications of GeFe2O4, it would help readers to understand GeFe2O4 more systematically.
Here are some specific suggestions:
1) Line 77---“Figure 2. Representation of the brunogeierite crystal structure: FeO6 octahedra (blue) and GeO4 tetrahedra (red).”
This picture cannot clearly exhibit the crystal structure to the reader. The resolution of the presented figures is not enough. The authors should adjust the display form to make it more legible.
2) In the Section “3. Syntheses”, the synthesis methods can be summarized in the table for comparison. As a review paper, more summary information should be included in the paper, (i.e, the advantages and disadvantages of each method and the applicable conditions).
3) Line 272 --- “The same piezochromic effect was verified for other Fe2+ minerals, and was due to the changes in the electronic band gap caused by crystal structure distortion and transition of octahedral Fe2+ from the low-spin (LS) to high-spin (HS) electronic state.” Please provide more references to prove the statement.
4) In section “5.2. Electrocatalys”, it states the possibility of application of GFO, and it is better to give a comparison with other catalysts, and research gaps.
